# In Silico and In Vitro Approach for Evaluation of the Anti-Inflammatory and Antioxidant Potential of Mygalin

**DOI:** 10.3390/ijms242317019

**Published:** 2023-11-30

**Authors:** Abraham Espinoza-Culupú, Nayara Del Santos, Mariella Farfán-López, Elizabeth Mendes, Pedro Ismael da Silva Junior, Monamaris Marques Borges

**Affiliations:** 1School of Nursing, Universidad Cesar Vallejo (UCV), Lima 15314, Peru; mfarfanlo@ucvvirtual.edu.pe; 2Bacteriology Laboratory, Butantan Institute, São Paulo 05585-000, Brazil; nayara.delsantos.esib@esib.butantan.gov.br (N.D.S.); elizabeth.mendes@butantan.gov.br (E.M.); 3Laboratory for Applied Toxinology (LETA), Butantan Institute, São Paulo 05585-000, Brazil; pedro.junior@butantan.gov.br

**Keywords:** acylpolyamine, mygalin, inflammation, antioxidant activity, molecular docking

## Abstract

There is a great interest in describing new molecules to be used as therapeutic targets in various diseases, particularly those that play a role in inflammatory responses and infection control. Mygalin is a synthetic analogue of spermidine, and previous studies have demonstrated its bactericidal effect against *Escherichia coli*, as well as its ability to modulate the inflammatory response of macrophages against lipopolysaccharide (LPS). However, the mechanisms through which mygalin regulates this inflammatory response remain poorly characterized. A set of platforms using molecular docking analysis was employed to analyze various properties of mygalin, including toxicity, biodistribution, absorption, and the prediction of its anti-inflammatory properties. In in vitro assays, we evaluated the potential of mygalin to interact with products of the inflammatory response, such as reactive oxygen species (ROS) and antioxidant activity, using the BMDM cell. The in silico analyses indicated that mygalin is not toxic, and can interact with proteins from the kinase group, and enzymes and receptors in eukaryotic cells. Molecular docking analysis showed interactions with key amino acid residues of COX-2, iNOS and 5-LOX enzymes. In vitro, assays demonstrated a significant reduction in the expression of iNOS and COX-2 induced by LPS, along with a decrease in the oxidative stress caused by the treatment with PMA, all without altering cell viability. Mygalin exhibited robust antioxidant activity in DPPH assays, regardless of the dose used, and inhibited heat-induced hemolysis. These studies suggest that mygalin holds promise for further investigation as a new molecule with anti-inflammatory and antioxidant properties.

## 1. Introduction

Natural products have been the main sources of chemical diversity, driving the discovery of new compounds with therapeutic potential. Polyamines play a crucial role in immune modulation and their analogues have shown promise in the treatment of diseases and infections [1].

Natural polyamines differ in the number of amine groups present in the molecule: putrescine has two amine groups, whereas spermidine and spermine contain three and four groups, respectively. This disparity imparts distinct effector characteristics to each polyamine [2]. For instance, spermine predominates in tumor cells and analogues based on its structure, inducing different effects in various cell types, underscoring the specificity of the analogues for different cell types [3]. This implies that these analogues can be utilized for different diseases, such as the polyamine analogues studied in cancer that affect tumor growth in cell culture [4] and act as potent bactericides [5].

Polyamines interact with nucleic acids and proteins, potentially influencing their structure and stability. Their binding to tRNA and mRNA can affect protein synthesis, growth, and differentiation [6,7]. Additionally, polyamines can modulate intracellular signals and immune functions, exhibiting anti-inflammatory and suppressive activity depending on the nature of the polyamine [8,9].

Mygalin is a bis-acylpolyamine N1, N8-bis (2,5-dihydroxybensoil) spermidine, with a molecular weight of 417 Da. It was originally isolated from hemocytes of the spider *Acanthoscurria gomesiana* [10]. Despite its structural homology with vertebrate spermidine, the fact that it originates from an invertebrate suggests that this molecule may possess unique characteristics. The objective of this study was to assess the structure of mygalin through a set of chemoinformatic tools to elucidate some of its properties including toxicity, biodistribution, absorption, and the prediction of inflammatory molecular targets in eukaryotic cells. In vitro tests were conducted to verify some of these targets in relation to anti-inflammatory and antioxidant activity.

## 2. Results

The discovery of a new candidate molecule for therapeutic applications demands swift and efficient optimization to identify the desired attributes and assess the likelihood of its success in practical use. Molecular docking has emerged as an increasingly important tool for expeditious and effective drug discovery. In this study, an in silico and in vitro analysis of mygalin was conducted to predict molecular properties, drug-likeness, lipophilicity, solubility, toxicity, ADME properties, and the target. This analysis was carried out using various chemoinformatic tools, including Molinspiration, SwissADME, SwissTargetPrediction and others.

### 2.1. In Silico Analysis

#### 2.1.1. Molecular Parameters

According to Lipinski’s rule of five, our molecule mygalin (Figure 1a) has a molecular weight of 417.46 which is less than 500 Da. The octanol/water partition coefficient (Log P) is <5 (Table 1). There are fewer than ten hydrogen bond acceptors (N and O), but it presents a violation, as there are more than 5 hydrogen bond donors (OH and NH). The boiled-egg diagram indicates that the compound mygalin has a low probability of passively passing through the blood–brain barrier and being absorbed by the gastrointestinal tract since it is located outside the yellow and white areas, respectively (Figure 1b). The bioavailability radar of the mygalin compound can be seen in Figure 1c, where the vertices represent parameters, and the pink regions represent the optimum range. However, for mygalin, only its flexibility falls outside the optimum range.

#### 2.1.2. Toxicity and ADME Prediction

In silico predictions were conducted to assess toxicity and ADME properties (Absorption, Distribution, Metabolism, and Excretion) with a log Kp (cm/s) value of −7.07. This specific characteristic of the mygalin molecule holds significant promise for its potential development as a future drug. These properties were calculated using various computational tools as outlined in Appendix A. The results from these tools suggest that mygalin is not toxic and carcinogenic according to AMES test predictions. This is supported by the prediction from MouseTox, which yielded an inactive and reliable result. Regarding its ability to penetrate the blood–brain barrier (BBB), mygalin exhibits low penetrability, as depicted in Figure 1b (as per QSAR Toolbox and SwissADME), or shows no penetration potential (as indicated by admetSAR).

#### 2.1.3. Target and Anti-Inflammatory Prediction 

In our analysis of potential molecular targets for mygalin, it was determined that our molecule can potentially interact with proteins belonging to the kinase group, Familia AG, membrane receptors, enzymes, and other categories (Figure 2a) within eukaryotic cells. This assessment was carried out using the SwissTargetPrediction Server and PharmaMapper, which leverage molecules for structural similarity to known compounds associated with protein targets. Results obtained from other servers including DDI-CPI and SEA are provided in the Appendix A. These results suggest that mygalin may have potential targets, such as proteins related to the nervous system, membrane receptors, and 5-LOX involved in the inflammatory process, among other proteins.

#### 2.1.4. Molecular Docking Analysis

Several enzymes play pivotal roles in the development of the inflammatory process. Among them are 5-LOX, involved in leukotriene biosynthesis, a group of lipid mediators derived from arachidonic acid, COX-2, responsible for catalyzing the formation of prostaglandins, and iNOS, which produces nitric oxide (NO) from L-arginine and plays a critical role in the immune defense against pathogens.

Consequently, we opted to assess, through molecular docking, the interaction of mygalin with these proteins that are integral to the inflammatory process.

The docking results are presented in Table 2 and Figure 3, Figure 4 and Figure 5. Mygalin exhibits interactions with key amino acid residues crucial for 5-LOX (ALA410 and LEU607); COX-2 (ARG121 and TYR356); iNOS (GLN257 and GLU371). These specific amino acids are known to interact with their respective substrates, such as arachidonic acid in the case of LOX or anti-inflammatory drugs in the cases COX-2 and iNOS.

### 2.2. In Vitro Analysis

#### 2.2.1. Effects of Mygalin on Nucleic Acid and the Protein Profile

It has been proven that polyamines bind to nucleic acids, potentially leading to their condensation [11,12,13]. This is, in part, due to the multiple positive charges these molecules carry, and their amino groups can associate with DNA or RNA [14,15]. Given that mygalin is an analogue of spermidine, we decided to investigate the impact of these molecules on nucleic acids after 24 h treatment with various concentrations of the compound. This model has been employed to elucidate the effects of drugs on DNA integrity in other cell types [16,17].

As shown in Figure 6a, the treatment with mygalin did not induce damage to eukaryotic cells. DNA and RNA remained unaltered without any observable changes. Furthermore, we assessed the protein profile of the cells after mygalin treatment by conducting a total protein extraction and visualizing the results on an SDS-PAGE acrylamide gel comparing them to cells that were not treated. Figure 6b reveals no significant differences between the cells treated with the drug and those left untreated, indicating that mygalin does not interfere with the protein synthesis of eukaryotic cells.

#### 2.2.2. Effect of Mygalin on the Expression of Genes iNOS and COX-2

The COX-2 and iNOS enzymes play crucial roles in regulating the inflammatory response, making the search for selective inhibitors of these molecules an area of significant interest. Consequently, we conducted an analysis of mygalin’s impact on bone marrow-derived macrophages and COX-2 and iNOS mRNA expression using RT-PCR after 6 h of LPS treatment (as depicted in Figure 7). Our observations revealed that the addition of mygalin (150 µM) led to a significant reduction in the expression of these enzymes when compared to treatment with LPS alone. This indicates that mygalin can interfere with the transcription of genes responsible for COX-2 and iNOS induced by LPS, thereby reducing the inflammatory effect produced.

### 2.3. Antioxidant Activity

#### 2.3.1. DPPH Radical Scavenging Activity of Mygalin

There is considerable interest in the research and development of antioxidant substances, particularly from natural products. Antioxidants have the ability to reduce free radicals, preventing the onset of various diseases caused by these radicals. One of the most employed techniques to detect the presence of antioxidant compounds is based on the elimination of the stable free radical 1,1-diphenyl-2-picrylhydrazyl (DPPH). We utilized this methodology to evaluate the antioxidant activity of mygalin. Figure 8 shows that mygalin at concentrations of 30–1000 µM showed high antioxidant activity (80%), differing from spermidine, with insignificant activity up to 500 µM (10%), and at the highest dose, 1000 µM, this activity was 40%. The reduction of free radicals by gentilic acid was greater than 40% only at concentrations above 500 µM. However, mygalin has ten times more capacity to neutralize free radicals compared to these compounds individually.

#### 2.3.2. Mygalin Reduced PMA-Induced Intracellular ROS in Macrophages

Natural polyamines, such as spermine and spermidine, are known to safeguard cells against the harmful effects of reactive oxygen species [18,19]. Given that mygalin is an acylpolyamine analogue of spermidine, we investigated whether this compound could impact the accumulation of ROS induced by PMA. Our findings revealed that mygalin exhibits antioxidative activity reducing PMA-induced oxidative stress in BMDM cells as shown in Figure 9.

#### 2.3.3. Heat-Induced Hemolysis

Erythrocyte survival is determined by the rate of metabolism, which is closely associated with the level of oxidative stress the cell undergoes. We assessed the impact of mygalin on the protection of sheep erythrocytes from hemolysis at various concentrations (ranging from 125 to 1000 µM). As illustrated in Figure 10, no notable hemolytic activity was observed even at the highest concentration of 1000 µM. This validates that mygalin provides protection to erythrocytes against heat-induced oxidative stress, likely by scavenging excess free radicals and preventing lipid peroxidation.

## 3. Discussion

The inflammatory process is highly complex, and some studies suggest that it may play a role in the initiation and progression of severe chronic diseases. This highlights the importance of exploring new drug options with anti-inflammatory properties that can interfere at various stages or factors within the inflammatory cascade to modulate inflammation.

Mygalin has emerged as a promising compound worthy of investigation due to its pharmacological mechanism, primarily stemming from its microbicidal activity against *E. coli*, resulting in cell membrane lysis and damage to bacterial DNA. This outcome can be partially attributed to the generation of oxidative stress and iron capture [10]. Interestingly, we have observed different behavior in eukaryotic cells, as mygalin does not induce cytotoxicity [20] nor inhibit the proliferation of ConA-activated splenocytes [21]. These characteristics were further elucidated in this study, where it was confirmed that mygalin did not cause damage to DNA and RNA nor lead to alterations in protein synthesis when bone marrow macrophages were treated with varying concentrations of mygalin ranging from 0–1000 µM (Figure 6). Given the complexity of the inflammatory response and the gaps in our understanding of various aspects of mygalin’s mechanism, ongoing research aims to uncover its pharmacological potential for therapeutic applications.

During the inflammatory response, an excessive release of pro-inflammatory immune mediators by tissue-resident cells can facilitate the migration of leukocytes from the bloodstream to the inflamed tissue, thereby coordinating local immunity. The persistent nature of this process can result in chronic inflammation, leading to tissue damage and impairing the functionality of the affected tissue [22].

Pro-inflammatory mediators such as cyclooxygenase 2 (COX-2) and inducible nitric oxide synthase (iNOS), in conjunction with the activation of the nuclear factor NF-kB, determine the extent of the developed inflammatory process [23]. Cyclooxygenase enzymes (COX) 1 and 2 are stimulated by pro-inflammatory cytokines, enhancing the synthesis of other molecules with inflammatory potential, including prostaglandins and thromboxanes, which act locally in the tissue, promoting a series of physiological changes [24]. On the other hand, iNOS is responsible for the generation of nitric oxide. Activation of this enzyme increases the inflammatory process and the control of innate immunity during acute and chronic inflammation, as well as the function of various cell types that assist in controlling microorganisms and pathogens caused by other diseases [25].

Recently, we demonstrated through molecular docking that mygalin can influence the regulation of the innate immune response by interacting with the TLR4/MD2 complex [20]. We confirmed that treatment of the Raw 246 cell line with varying doses of mygalin led to a reduction in the production of pro-inflammatory mediators induced by LPS, including the synthesis of cytokines TNF-α and IL-6 as well as the mRNA expression for COX-2, iNOS and NFkB [20]. In the present study, we expanded on these finding by utilizing primary bone marrow-derived macrophages (BMDM) pre-treated with 150 µM of mygalin, followed by LPS activation. This concentration was determined as sufficient to reduce mRNA expression for iNOS and COX-2, thus reinforcing prior observations made with phagocytic cell lines.

Macrophages play a critical role in orchestrating the immune response. They bridge the gap between innate and adaptive immunity through processes like antigen degradation, presentation, and cytokine production, all of which help regulate the inflammatory response. Marrow-derived macrophages originate from the myeloid lineage of the hematopoietic system and are distributed throughout various organs. These macrophages acquire distinct properties and phenotypes based on their location and the nature of the activation pathway [26]. As a result, they offer a valuable model for studying inflammation in vitro, closely resembling what occurs in the human body.

The utilization of in silico tools is of great value for predicting the activity of molecules, evaluating ADMET properties, elucidating their interaction with essential protein targets, and guiding investigations into the use of new compounds and drug repurposing [27,28,29]. The data derived from previous assays, as well as the current one utilizing this approach, align with the in vitro results, confirming that mygalin interacts with amino acid residues of 5-LOX, iNOS and COX-2 (Figure 3, Figure 4 and Figure 5, respectively); these are key targets for these enzymes. It is noteworthy that reported studies identify the same amino acids (Table 2) that interact with molecules known for their anti-inflammatory properties [30,31,32].

The in silico finding collectively demonstrate that mygalin is not cytotoxic and, as confirmed by the in vitro results, through molecular docking studies, interacts with the essential amino acids of the proteins 5-LOX, iNOS and COX-2, with is consistent with results reported by other researchers, confirming the anti-inflammatory nature of mygalin.

Free radicals including ROS and nitrogen species (RNS) are generated during normal cellular metabolism and in pathological processes [33]. These radicals play a role in the breakdown of phagocytized material, the regulation of cell growth, and control over intracellular signaling molecular mechanisms, contributing to the body’s innate immune defense against foreign agents [34,35]. However, an excess of these radicals during the inflammatory process can result in oxidation of biomolecules, particularly nucleic acids, lipids, proteins, and carbohydrates. This oxidative stress leads to tissue damage and disruption of normal physiological functions. To prevent and neutralize the excessive production of these radicals, the body employs a complex antioxidant defense system [33]. Depending on the nature and intensity of the inflammatory process, this defense system may not be able sufficiently to reverse the damage inflicted by oxidative stress, leading to chronic inflammation and degenerative diseases. The development of molecules with antioxidant properties is vital for preventing the onset of chronic inflammation and other diseases, benefiting the organism [36].

In this study, we demonstrated that mygalin reduced the generation of NO during macrophage activation with LPS. To explore the effect of this compound on oxidative stress generated by ROS, we examined the effect of mygalin on bone marrow macrophages treated with phorbol 12-myristate 13-acetate (PMA). The data presented in Figure 9 showed that mygalin reduced the generation of ROS induced by PMA. These findings suggest that mygalin may attenuate biochemical processes driven by oxidative stress in macrophages by quenching free radicals. To further substantiate these findings, we evaluated mygalin’s antioxidant activity through the DPPH assay which involves assessing color change (Figure 8), as well as its antihemolytic activity using erythrocytes (Figure 10). These experiments confirmed that mygalin possesses both antioxidant and anti-inflammatory activity.

The synthesis of mygalin involves the combination of spermidine and gentisic acid, resulting in a molecule with two acyl radicals [37]. Data from Figure 8 demonstrate that mygalin has high antioxidant activity, exceeding 80%, even at low concentrations (30 µM). The incorporation of acyl radicals plays a role in the antioxidant effect of mygalin, which contrasted with spermidine whose effect was only observed at high concentrations.

All cellular components are susceptible to the effects of oxidative stress. The cell membrane is a primary target due to lipid peroxidation, which results in structural change and altered membrane permeability. These changes influence ion exchange and the release of toxic products that can ultimately lead to cell death. The hemolysis test by heat demonstrated the effect of mygalin on protecting sheep erythrocytes from hemolysis induced by heat and was evaluated at different concentrations (125 to 1000 µM), Figure 10 shows that there was no significant hemolytic activity up to a concentration of 1000 µM, confirming that mygalin protected erythrocytes, in line with findings reported by other researchers [38,39] for molecules with anti-inflammatory and antioxidant properties.

## 4. Materials and Methods

### 4.1. Materials 

RPMI1640 medium; bovine serum albumin (BSA) 2,2-diphenyl-1-picrylhydrazyl (DPPH); LPS from *E. coli* serotype:0111:B4; EDTA; BHT (2,6-Di-tert-butyl-4methylphenol), Phorbol-12-Myristate-13-Acetate (PMA), DMSO, bovine albumin, agarose, penicillin, streptomycin were purchased from Sigma Aldrich Co. (St. Louis, MO, USA), CM-H2DCFDA was purchased from Thermo Fisher Scientific (Waltham, MA, USA).

Mygalin was synthesized and purified at the Center for Research on Toxins, Immune-Response and Cell Signaling (CeTICS—CEPID), Laboratory for Applied Toxinology (LETA)—Butantan Institute following the methodology descript in Espinoza-Culupú [10].

### 4.2. In Silico Studies

To predict pharmacokinetic parameters, targets, and anti-inflammatory activity, we used the chemical structure of mygalin in the form of canonical simplified molecular input line entry (SMILE): C1=CC(=C(C=C1O)C(=O)NCCCCNCCCNC(=O)C2=C(C=CC(=C2)O)O)O and to study the molecular docking a 3D structure obtained from Pubchem [40] was used in SDF format (https://pubchem.ncbi.nlm.nih.gov/compound/Mygalin (accessed on 3 March 2020)).

#### 4.2.1. Molecular Properties of Mygalin 

Molecular properties such as partition coefficient (Log P), topological polar surface area (TPSA), number of atoms, molecular weight, hydrogen bond donors and acceptors, rotatable bonds, and Lipinski’s “rule of five” were estimated using Molinspiration online server V 2018.10 (https://www.molinspiration.com/ (accessed on 5 March 2020)) and SwissADME.

#### 4.2.2. Toxicity and ADME Properties 

To assess toxicity and absorption, distribution, metabolism, and excretion (ADME) of mygalin, different approaches were carried out. The SMILE of mygalin was placed in QSAR Toolbox v4.4 [41] software for predicting and modelling toxicity using quantitative structure–activity relationship (QSAR). Additionally, it was analyzed using web servers such as AdmetSAR 2.0 [42], SwissADME online server (http://www.swissadme.ch/index.php/ (accessed on 20 April 2020)) [43], Lazar toxicity predictions [44], and MouseTox online server (https://www.enaloscloud.novamechanics.com/EnalosWebApps/MouseTox/ (accessed on 14 June 2020)) [45] to predict the cytotoxic effect of NIH/3T3.

#### 4.2.3. Targets and Anti-Inflammatory Prediction 

The canonical SMILE format of mygalin was submitted to servers DDI-CPI [46], SwissTargetPrediction online server (http://swisstargetprediction.ch/ (accessed on 20 April 2020)) [47], SEA online server (https://sea.bkslab.org/ accessed on 21 April 2020)) [48] and PharmMapper online server (https://lilab-ecust.cn/pharmmapper/index.html/ (accessed on 10 March 2023)) [49], to predict likely targets (receptors, enzymes, kinase etc.) based on similarity with other drugs. The target set was limited to mouse and human targets, and proteins related to the inflammatory process such as COX-2, iNOs, and Interleukins were selected.

#### 4.2.4. Molecular Docking 

The 3D crystal structures of iNOS (PDB: 3NQS) [50], COX-2 (PDB: 4PH9) [51], 5-LOX (PDB: 3V99) [52] in complex form with its inhibitors were obtained from Protein Data Bank (PDB) and 3D Structure from Mygalin was obtained from PubChem in SDF format and converted to PDB format with PyMol software (https://pymol.org/2/ (accessed on 16 September 2020)). The inhibitors were used to generate the grid then were subsequently removed from the structures. Molecular docking of Mygalin with 5-LOX, COX-2 and iNOS was conducted by Autodock vina v1.2 [53]; the parameters adopted were for 5-LOX (center_x = 18.39, center_y = −78.71 and center_z = −33.90; COX-2 (center_x = 13.00, center_y = 23.48 and center_z = 25.25); iNOS (center_x = 124.24, center_y = 115.29 and center_z = 35.35), with 0.375 Å spacing, 8 exhaustiveness and grid size X: 30, Y: 30 and Z: 30 Å for all. Binding modes receptor-ligands were visualized and analyzed using Discovery Studio Visualizer software 2020 [54].

### 4.3. In Vitro Studies

To demonstrate some of the results in silico, we used macrophage cultures and evaluated the expression of iNOS, COX-2.

#### 4.3.1. Animals

C57BL/6 mice were euthanized in a CO_2_ chamber, and then the bone marrow of the tibia and femur were extracted to obtain cells that were differentiated into bone marrow-derived macrophages (BMDM). The animals were obtained from the bioterium of the Faculty of Medicine and the Institute of Biomedical Sciences at USP. All procedures are in accordance with the approval of the Animal Ethics Committee of the Butantan Institute CEUA Protocol No. 5609301018.

#### 4.3.2. Isolation of BMDM and Cell Cultures

Cellular contents extracted from the femur and tibia to obtain the medullary cells that give rise to macrophages were used in our experiments using a standard protocol [55]. Briefly, the cells obtained were centrifuged at 1200 rpm and washed with PBS at 4 °C and cultured in RPMI 1640 medium (Gibco, Invitrogen Corporation, Waltham, MA, USA) supplemented with 10% fetal bovine serum (Gibco Invitrogen Corporation), 0.2 mM L-glutamine, 50 IU/mL penicillin and 50 μg/mL streptomycin, 1 mM sodium pyruvate, 1 mM non-essential amino acids, 50 μM β-mercaptoethanol and 25% of the L929 cell supernatant (*v*/*v*) in Petri dishes for cell culture for 7 days. On the third and fifth day, half of the supernatant was replaced with a new medium and the cells were incubated under the same conditions. On the seventh day, the differentiated cells were washed with PBS pH 7.2 to remove non-adherent cells and the adherent cells were scraped and plated at 1.5 × 10^6^ cells/500 μL/well in 24-well culture plates and maintained for 18 h in the same conditions. After this incubation period, the medium was discarded, and the cells were washed with PBS and cultured in RPMI medium. Cell line J774A.1 (ATCC) was reactivated in RPMI medium supplemented with 20% Bovine Fetal Serum (SFB), at 37 °C with 5% CO_2_, the next day, placed in a new medium with 10% SFB and gentamicin (25 μg/mL) (complete RPMI). The macrophage cells were maintained in serial cultures in the incubator at 5% CO_2_ and 37 °C. For the assays, they were plated at a similar condition of BMDM in 24-well plates (Costar^®^, Cambridge, MA, USA).

#### 4.3.3. Activation of Macrophages, and Inflammation

Cell line J774A.1 were cultured in 24-well plates using RPMI medium (complete) and exposed to mygalin to assess DNA, RNA, and protein damage through electrophoresis analysis. In another group of assays, BMDM cells were pre-treated with mygalin (150 μM) for 1 h before the addition of LPS 100 ng/mL; after 6 h of stimulation the cells were collected for total RNA extraction to evaluate the gene expression of iNOS and COX-2.

#### 4.3.4. RNA Extraction and RT-PCR

RNA extraction and RT-PCR followed the methodology reported in [20]. Cells were washed with PBS (500 μL per well) to discard the residue of the culture medium, QiAzol Lysis Reagent (Qiagen^®^, Hilden, Germany) was added per well and the RNA purification process was conducted using Direct-Zol Kit (ZYMO RESEARCH, Irvine, CA, USA). RNA was converted to cDNA using a High-Capacity RNA-to-cDNA Kit (Applied Biosystems^TM^, Waltham, MA, USA), and cDNAs were used as a template for the amplification of the genes iNOS and COX-2. RT-PCR Products were separated by electrophoresis on 1.5% agarose gels stained with gelRed. The quantifications of the densities of the PCR products were made using the Image J program v 1.7.1 (http://imagej.nih.gov/ij/ (accessed on 26 February 2021)).

### 4.4. In Vitro Antioxidant and Anti-Inflammatory Activity of Mygalin

#### 4.4.1. DPPH Radical Scavenging Assay

The antioxidant activity of mygalin (0–1000 µM) was evaluated by DPPH free radical, using the method described by Braca et al. [56] with modifications by Wang et al. [57]; 20 µL of each sample in methanol by triplicate was added to 180 μL of DPPH solution (0.2 mM in methanol) using plates of 96-well costar, then gently shaken and left in the plate for 30 min at 37 °C in the dark. The absorbance of the DPPH was measured at 515 nm and percentage of DPPH was calculated using the following formula:% antioxidant activity = ((Abs control − Abs sample)/Abs control) × 100

Spermidine was used as a positive control.

#### 4.4.2. Measurement of Intracellular Reactive Oxygen Species (ROS)

BMDM (10^5^ cells/well) were cultured in black COSTAR^®^ 96-well microplate and incubated at 37 °C for 24 h in RPMI medium, after which medium was removed and pretreated with Mygalin (300 µM) for 30 min in HSBB solution. The cells were stimulated with PMA (200 nM) for 30 min, followed by the addition of 10 µM CM-H_2_DCFDA for 30 min more [58]. The plates were incubated in the absence of light, and the fluorescence was measured using a PerkinElmer Victor 3^TM^ 1420, Waltham, MA, USA, Multilabel Counter Fluorometer with 485/535 nm excitation/emission wavelength.

### 4.5. In Vitro Anti-Inflammatory Activity

#### Heat-Induced Hemolysis

A suspension of erythrocytes was prepared following the methodology of Gunathilake et al. [59], and blood was washed three times with saline solution (0.9%). Then Erythrocytes suspension was reconstituted as a 10% (*v*/*v*) in saline solution. Then 50 µL of blood cell suspension and 50 µL of mygalin (0–1000 µM) were mixed with 2.95 mL saline solution (pH 7.4), and the mixture was mixed gently and incubated at 54 °C for 20 min in a water bath, then centrifuged at 2500 rpm for 3 min and the absorbance of the supernatant was measured at 550 nm using a UV/VIS spectrometer. The level of hemolysis was calculated using the following formula:% inhibition of hemolysis = ((Abs control − Abs sample)/Abs control)) × 100

### 4.6. Statistical Analysis

The data was expressed as mean ± SEM of at least three independent tests, the results were analyzed using Student’s *t*-test or ANOVA, and the difference between groups determined by the Tukey–Kramer or Dunnett test for Multiple comparisons using the GraphPad Prism 7 program (Graph Pad, San Diego, CA, USA). The data were considered statistically significant with a *p* < 0.05.

## 5. Conclusions

In conclusion, the data accumulated from in silico and in vitro assays, along with the use of phagocytic cells as a model, among other tests, strongly suggest that mygalin is a promising molecule. It has the potential to influence the innate immune response by modulating inflammatory activity through the production of immune mediators with anti-inflammatory properties, while also exhibiting antioxidant activity. Further investigations in this direction should be explored.

## Figures and Tables

**Figure 1 ijms-24-17019-f001:**
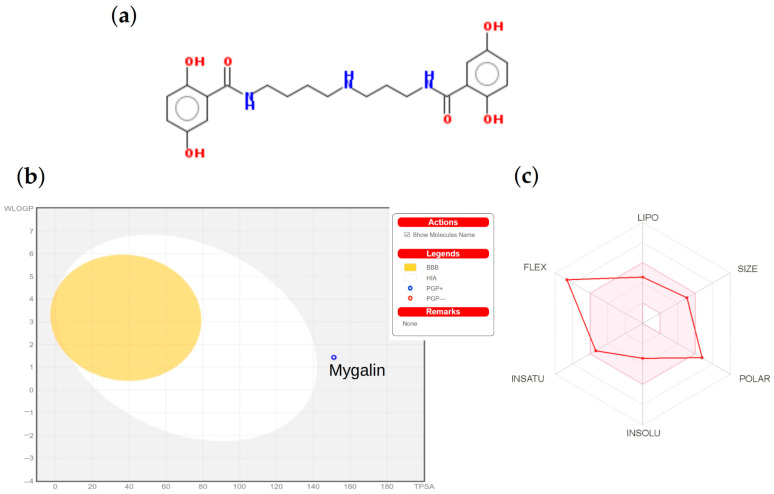
(**a**) Chemical structure of mygalin, oxygen in red and nitrogen in blue color (**b**) Predicted BOILED-Egg diagram (**c**) Bioavailability radar from SwissADME web tool; in this last, the pink area shows the optimal range of particular properties. LIPO (lipophilicity), SIZE (molecular weight), POLAR (polarity), INSOLU (insolubility in water), INSATU (unsaturation), and FLEX (flexibility).

**Figure 2 ijms-24-17019-f002:**
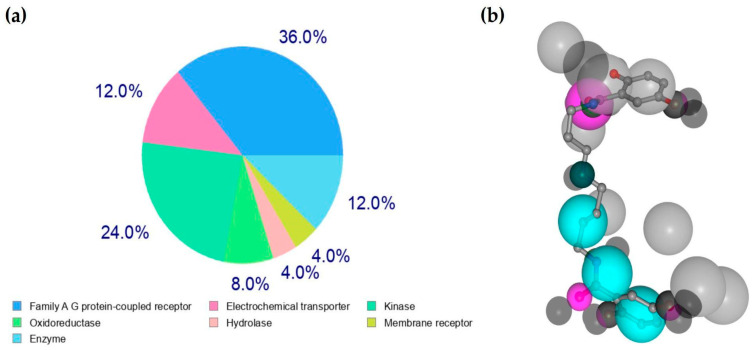
(**a**) Protein target class distribution for mygalin using SwisTargetPrediction with N° 25; (**b**) Pharmacophore model of mygalin to iNOS protein. Pharmacophore colour schemes are indicated by colour: hydrophobic is cyan and acceptor is magenta.

**Figure 3 ijms-24-17019-f003:**
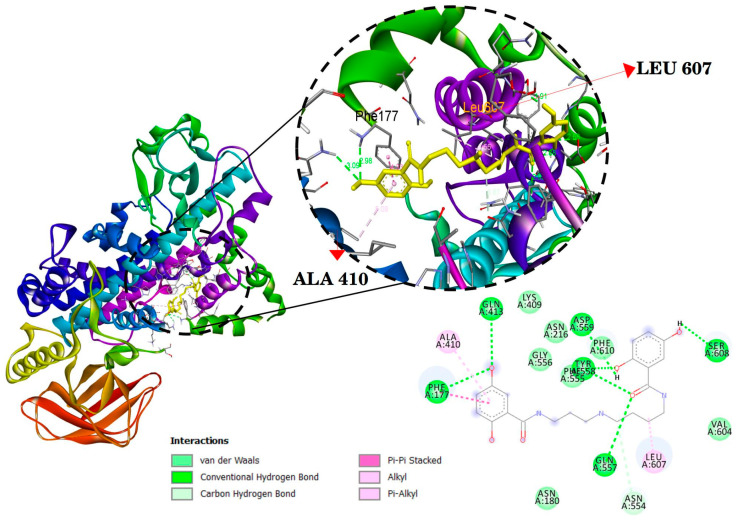
Molecular docking of mygalin (Yellow) and its interactions with the active site the crystal structure of 5-Lipoxygenase (5-LOX) (PDB code: 3V99). Binding sites with the interactions are shown in amino acid residues: PHE177, ALA410, GLN413, ASN554, PHE555, GLN557, TYR558, ASP559, LEU607 and SER608. Residues involved in van der Waals interactions, hydrogen bonding, carbon-hydrogen, and Pi-alkyl are represented in different color indicated in the set.

**Figure 4 ijms-24-17019-f004:**
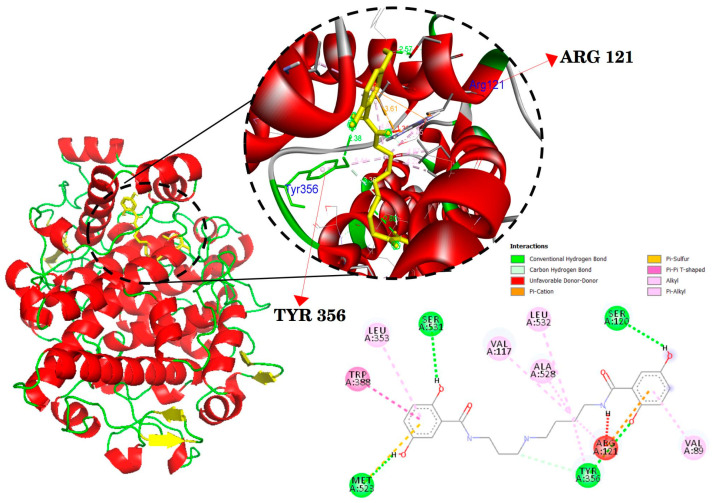
Molecular docking of mygalin (Yellow) and its interactions with the active site with the crystal structure of cyclooxygenase 2 (COX-2) (PDB code: 4PH9). Binding sites with the interactions are shown in amino acid residues: VAL89, VAL117, SER120, ARG121, LEU353, TYR356, TRP388, MET523, ALA528, SER531 and LEU532. Residues involved in van der Waals interactions, hydrogen bonding, and Pi-Anion are represented in different color indicated in the set.

**Figure 5 ijms-24-17019-f005:**
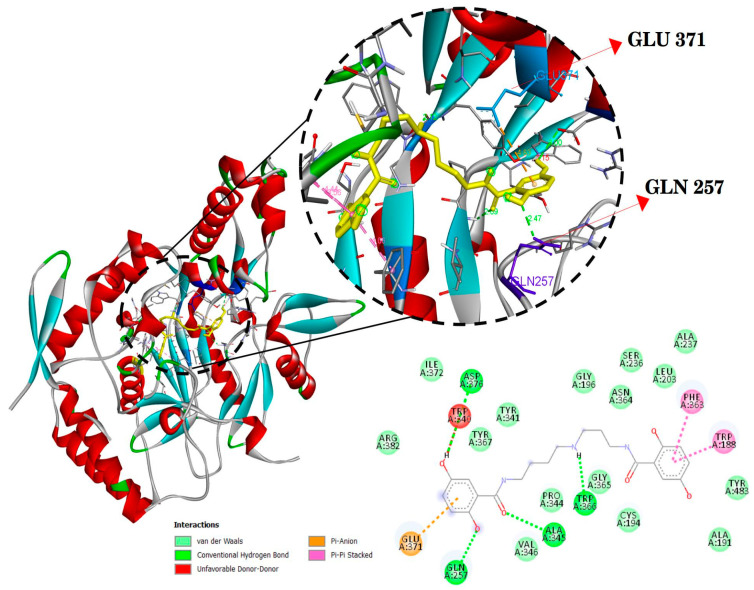
Molecular docking of mygalin (Yellow) and its interactions with the active site the crystal structure of iNOS (PDB code: 3NQS). Binding sites with the interactions are shown in amino acid residues: TRP188, GLN257, TRP340, ALA345, PHE363, TRP366, GLU371 and ASP376. Residues involved in van der Waals interactions, hydrogen bonding, and Pi-Anion are represented in different color indicated in the set.

**Figure 6 ijms-24-17019-f006:**
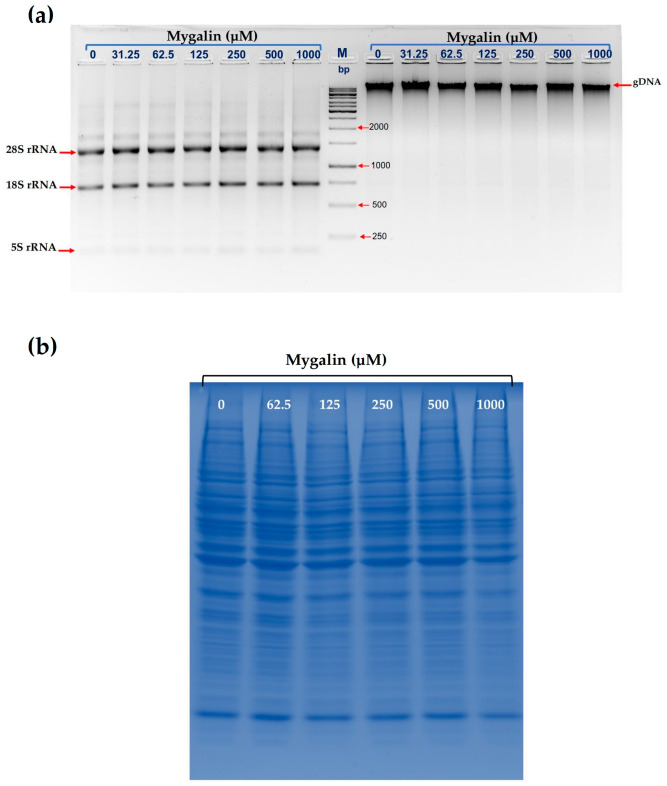
Nucleic acid and protein profile of macrophage cell J774 treatment with mygalin. (**a**) 1.5 × 10^6^ cells/well were placed in a 24-well plate and incubated with mygalin (0–1000 µM) for 24 h. Cells were collected and acid nucleic was isolated from each treatment. RNA (**Left**) and DNA (**Right**) were analyzed on 1.2% agarose gel. M: Ladder Gene Ruler 1 Kb. (**b**) Cells were lysed with RIPA buffer and the proteins visualized on SDS-PAGE acrylamide gel and stained with Coomassie blue.

**Figure 7 ijms-24-17019-f007:**
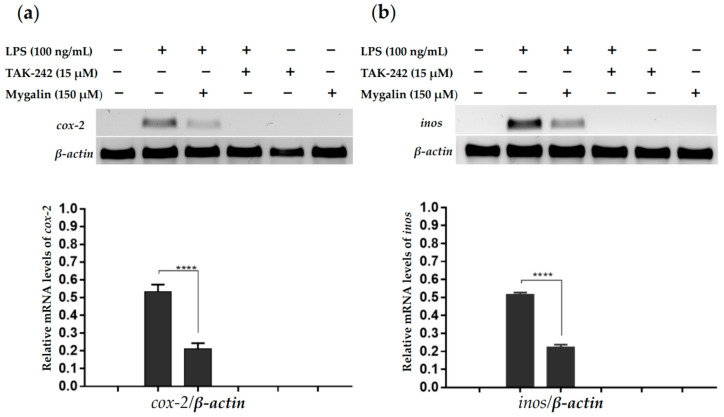
Effect of mygalin on the expression of *cox-2* (**a**) and *inos* (**b**) genes in bone marrow-derived macrophages; (1.5 × 10^6^ cells/well) was treated with mygalin (150 µM) for 1 h and incubated with LPS (100 ng/mL) or TAK-242 (15 µM) for 6 h. mRNA levels were analyzed by RT-PCR and normalized to *β-actin*. (N = 3 trials). The RT-PCR products were visualized on a 1.5% agarose gel. The bar represents the mean ± SEM of three independent experiments (**** *p* < 0.0001).

**Figure 8 ijms-24-17019-f008:**
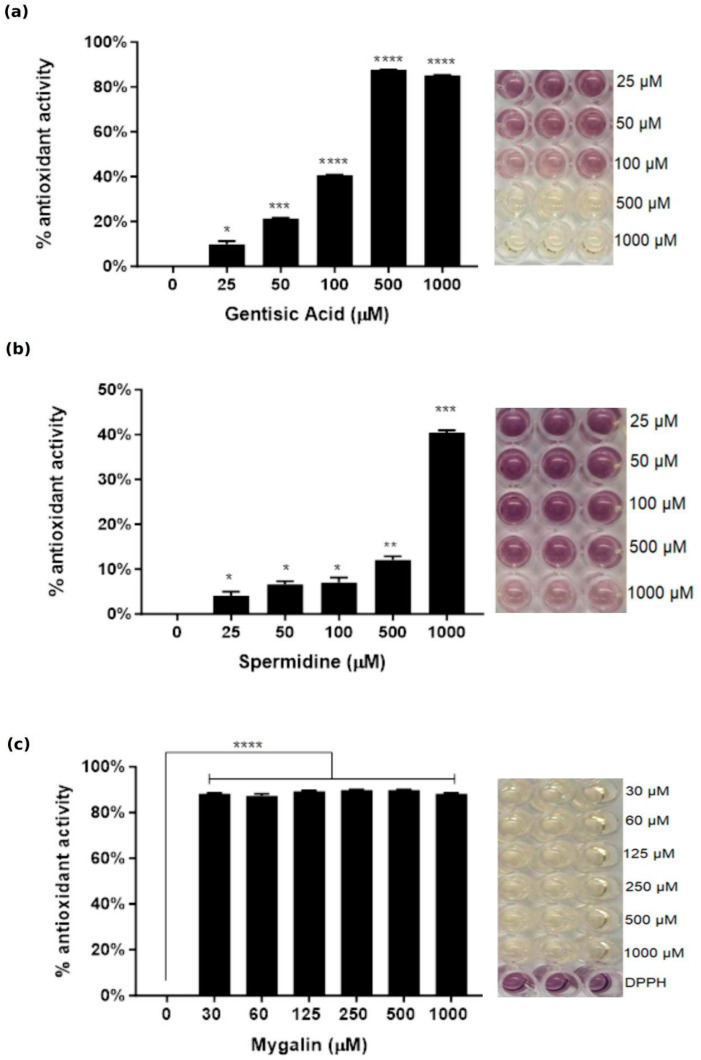
Mygalin antioxidant activity DPPH method. One solution containing 20 µL sample plus 180 µL of DPPH (0.2 mM) was gently shaken and incubated for 30 min at 37 °C in the dark. The absorbance of the DPPH was measured at 515 nm. (**a**) gentisic acid, (**b**) spermidine and (**c**) mygalin. Spermidine was used as a positive control. The conversion of purple to yellow colour indicates the presence of antioxidant activity. The bar represents the mean ± SEM of three independent experiments (* *p* < 0.05, ** *p* < 0.01, *** *p* < 0.001 and **** *p* < 0.0001).

**Figure 9 ijms-24-17019-f009:**
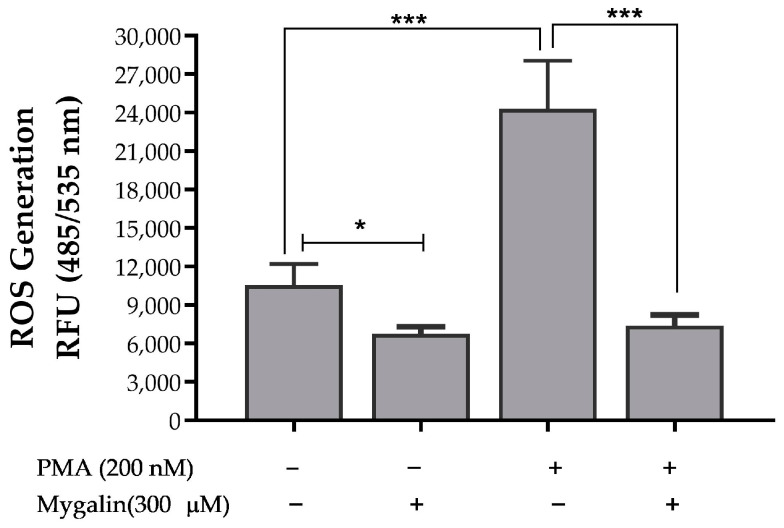
Mygalin inhibits PMA-induced oxidative stress. BMDM (10^5^ cells/well) were pretreated for 30 min with mygalin (300 µM) and subsequent incubation with PMA (200 nM) for 30 min. The bar graph (y-axis) represents the relative fluorescence unit (RFU) of intracellular ROS levels and the x-axis represents the activation. Data represent mean ± S.E.M, of 3 individual experiments. * *p* < 0.05 and *** *p* < 0.001.

**Figure 10 ijms-24-17019-f010:**
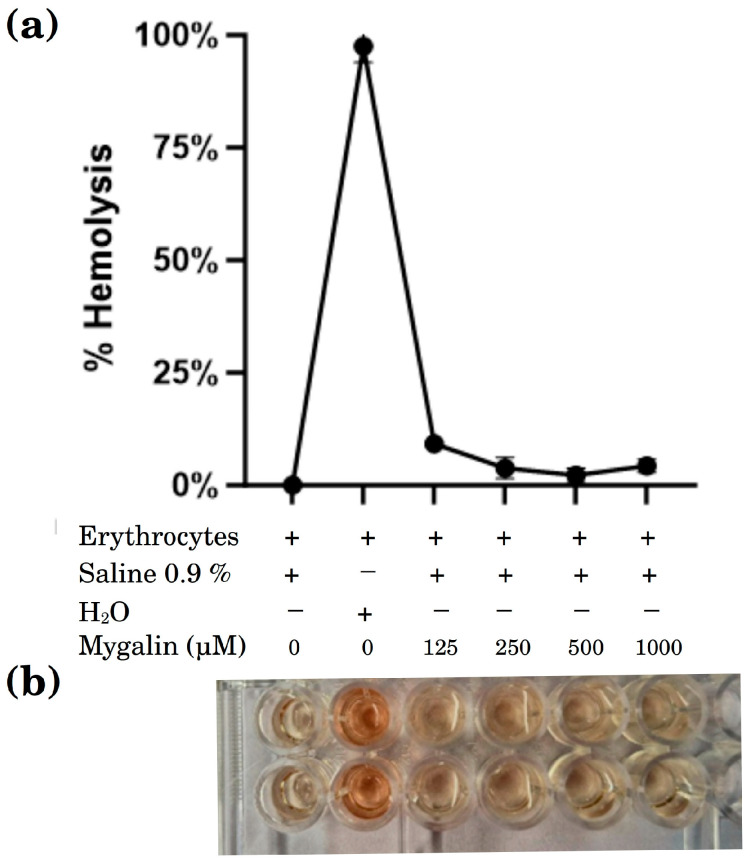
Effect of mygalin on heat-induced hemolysis. (**a**) 50 µL of erythrocyte suspension and 50 µL mygalin (125–1000 µM) were added to 1 mL of saline solution. After incubation for 20 min to 54 °C and subsequent centrifugation, the absorbance of the supernatant was measured at 550 nm. (**b**) The level of hemolysis was calculated using the formula (% hemolysis = ((Control Abs-sample Abs)/Control Abs)) × 100). Data show the mean ± SEM of two independent experiments.

**Table 1 ijms-24-17019-t001:** Molecular properties of Mygalin.

Molecular Properties	Molinspiration	SwissADME
Molecular weight	417.46	417.46
Heavy atoms	30	30
Aromatic heavy atoms	-	12
Rotatable bonds	-	13
H-bond acceptors	9	7
H-bond donors	7	7
Molar refractivity	-	111.69
TPSA [Ǻ2]	151.13	151.15
Consensus log P 0/w or miLogP	2.17	1.77
Lipinski violation	1	1
Solubility Log S	-	−3.44

**Table 2 ijms-24-17019-t002:** Docking energy and binding-site residues of iNOS, COX-2, and 5-LOX with mygalin using Autodock vina.

Protein	Docking Energy (kcal/mol)	Binding-Site Residues
5-LOX	−7.70	PHE177, ALA410, GLN413, ASN554, HE555, GLN557, TYR558, ASP559, LEU607 and SER608.
COX-2	−7.40	VAL89, VAL117, SER120, ARG121, LEU353, TYR356, TRP388, MET523, ALA528, SER531 and LEU532.
iNOS	−8.00	TRP188, GLN257, TRP340, ALA345, PHE363, TRP366, GLU371 and ASP376

## Data Availability

Data is contained within the article and Appendix A.

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
