# Peer review of "In Silico and In Vitro Approach for Evaluation of the Anti-Inflammatory and Antioxidant Potential of Mygalin"

_ijms, 2023, doi:10.3390/ijms242317019_

Round 1
Reviewer 1 Report
Comments and Suggestions for Authors
This is an interesting paper looking at the properties of mygalin. The authors are thorough in their analyses, demonstrating that it can bind inflammatory proteins directly as well as inhibit their RNA expression. I have a few comments for consideration.
1. It is curious that the LPS induction of genes is inhibited. Could this be due to decreased TLR4 expression?
2. Is the inhibition seen if IFN-g is also included in the stimulation? (LPS + IFN-g is a commonly used combination in studies focusing on inflammation).
3. Is there a negative control for the stimulation (i.e. are all genes decreased in the presence of mygalin)?
4. Are the effects reversible? If so, in what time frame?
5. Can mygalin reduce ROS levels if added after PMA?
6. Does mygalin block signaling through other TLR receptors?
Author Response
Dear Reviewer
Thank you very much for taking the time to review this manuscript. We appreciate your comments and observations on the submitted manuscript. We think they are relevant to improve the article. Please find the detailed responses below and the corresponding revisions/corrections are in color blue in the attachment
The English grammar was revised and changes can be seen in blue
Questions:
1- It is curious that the LPS induction of genes is inhibited. Could this be due to decreased TLR4 expression?
REPLY: Thank you for your observation.
TLRS activation is complex and determined by various mechanisms, including the expression of the receiver itself, its location, adaptive molecules and signal transduction pathways. The influence of mygalin on the TLR4 expression profile is being analysed as well as some other aspects mentioned and will be presented soon.
2- Is the inhibition seen if IFN-g is also included in the stimulation? (LPS + IFN-g is a commonly used combination in studies focusing on inflammation).
REPLY: Thank you for your observation.
Tests performed and ongoing showed that the activation of macrophages with IFN-γ partially attenuates mygalin's suppressive action on LPS-induced response for some immune mediators (manuscript in preparation).
3.- Is there a negative control for the stimulation (i.e. are all genes decreased in the presence of mygalin)?
REPLY: Thank you for your observation.
The negative control is the cells without treatment. In Figure 7, it would correspond to the first lane for both genes evaluated (COX-2 and iNOS).
4.-Are the effects reversible? If so, in what time frame?
REPLY: Thank you for your observation
These essays have not yet been performed.
5.- Can mygalin reduce ROS levels if added after PMA?
REPLY: Thank you for your observation
These essays have not yet been performed.
6.- Does mygalin block signaling through other TLR receptors?
REPLY: Thank you for your observation
These essays are being performed (manuscript in preparation).
Reviewer 2 Report
Comments and Suggestions for Authors
Some remarks:
The authors should add a list of abbreviations to the manuscript.
They could improve the study by comparing the in-silico toxicity with the in vitro toxicity.
As for the figures 3-5 of the manuscript, the labels are not easy to read.
Comments on the Quality of English LanguageThe manuscript contains some spelling errors, too.
Author Response
Dear Reviewer
Thank you very much for taking the time to review this manuscript. We appreciate your comments and observations on the submitted manuscript. We think they are relevant to improve the article. Please find the detailed responses below and the corresponding revisions/corrections are in color blue in the attachment
The English grammar was revised and changes can be seen in blue
Questions:
1.-The authors should add a list of abbreviations to the manuscript.
Reply: Thank you for your observation and suggestion.
we followed the Instructions for authors. "abbreviations are shown in the text"
2.- They could improve the study by comparing the in-silico toxicity with the in vitro toxicity.
Reply: Thank you for your observation
Was added lines 302-303
“The in silico finding collectively demonstrate that mygalin is not cytotoxic and confirmed by the in vitro results”
3.- As for the figures 3-5 of the manuscript, the labels are not easy to read.
Reply: Thank you for your observation
Figures 3, 4 and 5 were changed, with red arrows indicating the key amino acids highlighted in black.

Reviewer 3 Report
Comments and Suggestions for Authors
Dear authors
I read your article very carefully.
My recommendation will be to publish with major revisions, based on the observations formulated below:
- I recommend the authors to introduce a sentence to explain the diagram of the boiled egg.
- The kinases assumed as potential targets, are they general or are there certain specific kinases?
- In the sentence regarding the way of tying the migalina, I think it would be advisable for the authors to cover the statements made (lines 130-134) with bibliographic indexes. Otherwise, the statements are too general. Why not the authors and spermidine in the docking studies? For comparison.
- Is the dose used by migalin for the action on COX and iNOS similar to that used to evaluate the effect on nucleic acids and proteins?
- For me, the expression of the results on DPPH (although I know the method well) is not very clear. Maybe the authors could rewrite this part in a clearer way.
- In figure 9 what do the columns represent. There is no legend on the x-axis.
- Figure 10 again is unclear.
- Lines 257-258 - bibliography!
In conclusion, most observations are related to the presentation of the results, the figures being insufficiently clear.
Also as a general recommendation, I think that the authors should check, with the help of a native speaker, the English language used.
Comments on the Quality of English LanguageAs a general recommendation, I think that the authors should check, with the help of a native speaker, the English language used.
Author Response
Dear Reviewer
Thank you very much for taking the time to review this manuscript. We appreciate your comments and observations on the submitted manuscript. We think they are relevant to improve the article. Please find the detailed responses below and the corresponding revisions/corrections are in color blue in the attachment
The English grammar was revised and changes can be seen in blue
1.- I recommend the authors to introduce a sentence to explain the diagram of the boiled egg.
Reply: Thank you for your analysis and suggestions
“The boiled-egg diagram indicated that the compound mygalin has a low probability of passively passing through the blood-brain barrier and being absorbed by the gastrointestinal tract since it is located outside the yellow and white area respectively (Figure 1b)” was added line 76-79
2.- The kinases assumed as potential targets, are they general or are there certain specific kinases?
Reply: Thank you for your analysis
In another study, our research group is conducting tests to investigate the action of mygalin in kinase phosphorylation and we can soon respond.
3.- In the sentence regarding the way of tying the migalina, I think it would be advisable for the authors to cover the statements made (lines 130-134) with bibliographic indexes. Otherwise, the statements are too general. Why not the authors and spermidine in the docking studies? For comparison.
Reply: Thank you for your analysis
Mygalin molecular docking tests with IFN-γ and spermidine are underway on another project and will be published soon.
4.- Is the dose used by migalin for the action on COX and iNOS similar to that used to evaluate the effect on nucleic acids and proteins?
Reply: Thank you for your analysis and suggestions
The dose used to evaluate the effect of mygalin on nucleic acids and protein was 30 to 1000 µM (Fig.7). In the COX and iNOS tests it was 150 µM (Fig.9)
5.- For me, the expression of the results on DPPH (although I know the method well) is not very clear. Maybe the authors could rewrite this part in a clearer way.
Reply: Thank you for your analysis and suggestions
The text was corrected, the changes are in blue
6.- In figure 9 what do the columns represent. There is no legend on the x-axis.
Reply: Thank you for your analysis
The Y-axis represents the relative fluorescence units (RFU) and the X-axis the PMA and mygalin stimulus, With stimulus (+), No stimulus (-)
The text was corrected, the changes are in blue
7.- Figure 10 again is unclear.
Reply: Thank you for your analysis
the legend was modified: figure 10. Effect of mygalin on heat-induced hemolysis. a) 50 µL of erythrocyte suspension and 50 µL mygalin (125-1000µM) were added to 1 mL of saline solution. After incubation for 20 min to 54° C and subsequent centrifugation, the absorbance of the supernatant was measured at 550 nm, b) The level of hemolysis was calculated using the formula (% hemolysis = ((Control Abs-sample Abs)/Control Abs)) *100). Data show the mean ± SEM of two independent experiments.
8.- Lines 257-258 - bibliography!
Reply: Thank you for your observation
Added reference line 270
“Tak, P.P.; Firestein, G.S. NF-κB: A Key Role in Inflammatory Diseases. J. Clin. Invest. 1 2001, 107, 7–11, doi:10.1172/jci11830”
And reference line 274
Minghetti, L. Cyclooxygenase-2 (COX-2) in Inflammatory and Degenerative Brain Diseases. J. Neuropathol. Exp. Neurol. 2004, 63, 901–910, doi:10.1093/jnen/63.9.901.
Reviewer 4 Report
Comments and Suggestions for Authors
Please see attached

Double check for typing errors
Author Response
Dear Reviewer
Thank you very much for taking the time to review this manuscript. We appreciate your comments and observations on the submitted manuscript. We think they are relevant to improve the article. Please find the detailed responses below and the corresponding revisions/corrections are in color blue in the attachment
The English grammar was revised and changes can be seen in blue
Questions:
1.- Line 42-43 - another important example is a study of spermine/spermidine analogs and synthetic MSO polyamine compounds as inhibitors of Glutamine synthetase - like enzymes for potentially treatment of bacterial infections, e. g. tuberculosis (https://doi.org/10.1093/femsle/fnad096 ).
Reply: Thank you for your observation
We were unable to obtain the recommended reference. Please if you can send it to add your recommendation for more was added other references related to analogues of polyamines,
“and potential bactericidal” [5].
Chen, D.; Cadelis, M.M.; Rouvier, F.; Troia, T.; Edmeades, L.R.; Fraser, K.; Gill, E.S.; Bourguet-Kondracki, M.-L.; Brunel, J.M.;Copp, B.R. Α,ω-Diacyl-Substituted Analogues of Natural and Unnatural Polyamines: Identification of Potent Bactericides That Selectively Target Bacterial Membranes. Int. J. Mol. Sci. 2023, 24, 5882, doi:10.3390/ijms24065882
2- Line 47-48 - suggest to add that polyamines play a great role in bacterial survival in specific ecological niches, such as human macrophages (e.g. Mycobacterium tuberculosis)
Reply: Thank you for your observation
Research on other pathogenic bacteria, such as biofilm formers, is being carried out in our laboratory, to evaluate the role of polyamines in bacterial internalization.
3- Line 69-70 - please add references to the respective tools mentioned to be used in the study
Reply: Thank you for your observation
The references used were cited in item 4.2.2
4- Line 131 - are these key amino acid resides located in the catalytic pocket/active site of the enzyme since they are involved in substrate binding? Suggest to clarify.
Reply: Thank you for your observation
Line 136-140 was modified
“The docking results are presented in Table 2 and Figures 3, 4, and 5. Mygalin exhibits interactions with key amino acid residues crucial for 5-LOX (ALA410 and LEU607); COX-2 (ARG121 and TYR356); iNOS (GLN257 and GLU371). These specific amino acids are known to interact with their respective substrates, such as arachidonic acid in the case of LOX or anti-inflammatory drugs in the cases COX-2 and iNOS”
This key amino acid residues are part of catalitic site
5- Line 227-228 - please add citation to the mentioned studies
Reply: Thank you for your observation
References are mentioned on line 341 “Figure 10 shows that there was no significant hemolytic activity up to a concentration of 1000 µM, confirming that mygalin protected erythrocytes, in line with findings reported by others researchers [38, 39]”
6.-Line 322 - there is a double space between [36.37] and "in molecules"
Reply: Thank you for your observation
line 322 space was deleted
7- Line 372 - reference for "Discovery Studio Visualizer software. " Line 577 - check formatting
Reply: Thank you for your observation
Was added reference [54] line 391
“Biovia, D.S. Discovery Studio Visualizer 2020. Release v20 1.0.19295; Dassault Systèmes: San Diego, CA, USA, 2016”
Round 2
Reviewer 1 Report
Comments and Suggestions for Authors
No other comments
Reviewer 2 Report
Comments and Suggestions for Authors
No comments.
Reviewer 3 Report
Comments and Suggestions for Authors
Dear authors,
I reread your study and I appreciate favorably the fact that you took into account the observations made. I think that the article won in quality.
At this moment, I believe that the article can be accepted for publication as it is.
Reviewer 4 Report
Comments and Suggestions for Authors
Authors addressed all reviewer comments